# Align Human Camouflaged Perception: Visual Refocus Reinforcement Fine-Tuning

## Abstract

Current multi-modal models exhibit a notable misalignment with the human visual system when identifying objects that are visually assimilated into the background. Our observations reveal that these multi-modal models cannot distinguish concealed objects, demonstrating their inability to emulate human cognitive processes that effectively utilize foreground-background similarity principles for visual analysis. To analyze this hidden human-model visual thinking discrepancy, we build a visual system that mimics human visual camouflaged perception to progressively and iteratively 'refocus' concealed visual content. The refocus is a progressive guidance mechanism enabling models to logically localize objects in visual images through stepwise reasoning. The localization process of concealed objects requires hierarchical attention shifting with dynamic adjustment and refinement of prior cognitive knowledge. In this paper, we propose a visual refocus reinforcement framework (VRRF) via the policy optimization algorithm to encourage multi-modal models to think and refocus more before answering, and achieve excellent reasoning abilities to align human camouflaged perception systems. Our extensive experiments on camouflaged perception successfully demonstrate the emergence of visual refocus phenomena, characterized by multiple reasoning tokens and dynamic adjustment of the detection boxes. Besides, experimental results on both camouflaged object classification and detection tasks exhibit significantly superior performance compared to Supervised Fine-Tuning (SFT) baselines. Furthermore, our visual refocus system surpasses the human camouflaged perception in user study where participants are required to identify the target object. We hope that this work will provide valuable insights for advancing future research in multimodal model development. The implementation code and datasets will be made publicly.

## 1 Introduction

Recent breakthroughs in applying reinforcement learning (RL) to large language models (LLMs) have yielded significant advancements. As demonstrated by OpenAI-O1 (Jaech et al., 2024), reinforcement fine-tuning (RFT) strategies exhibit substantial potential for enhancing LLMs' complex reasoning abilities. Furthermore, DeepSeek-R1-Zero (Guo et al., 2025a) established that rule-based reward systems can effectively leverage RL to unlock superior reasoning and cognitive capabilities in language models, even without extensive supervised fine-tuning.

Recent research has increasingly sought to replicate the success of DeepSeek-R1 in multimodal large language models (MLLMs). Notably, Virgo (Du et al., 2025) employed knowledge distillation from open-source reasoning models including DeepSeek-R1 (Guo et al., 2025a) to enhance visual reasoning capabilities. However, the dominant research direction (Zhou et al., 2025a; Liu et al., 2025a; Zhan et al., 2025; Deng et al., 2025a; Peng et al., 2025; Liu et al., 2025b; Yang et al., 2025a; Zhang et al., 2025; Deng et al., 2025b) prioritizes direct implementation of DeepSeek-R1's Group Relative Policy Optimization (GRPO) alongside its rule-based reward system to enable visual reasoning in MLLMs. This approach has primarily focused on improving performance in STEM tasks, particularly those involving mathematical reasoning and counting challenges. A fundamental driver of this success lies in the rule-based reward system's inherent applicability to tasks with deterministic ground-truth solutions, which enables both stable and interpretable reward signaling. In the visual domain, many visually intensive tasks demand deeper cognitive processing and greater attention to visual information than reasoning with LLMs.

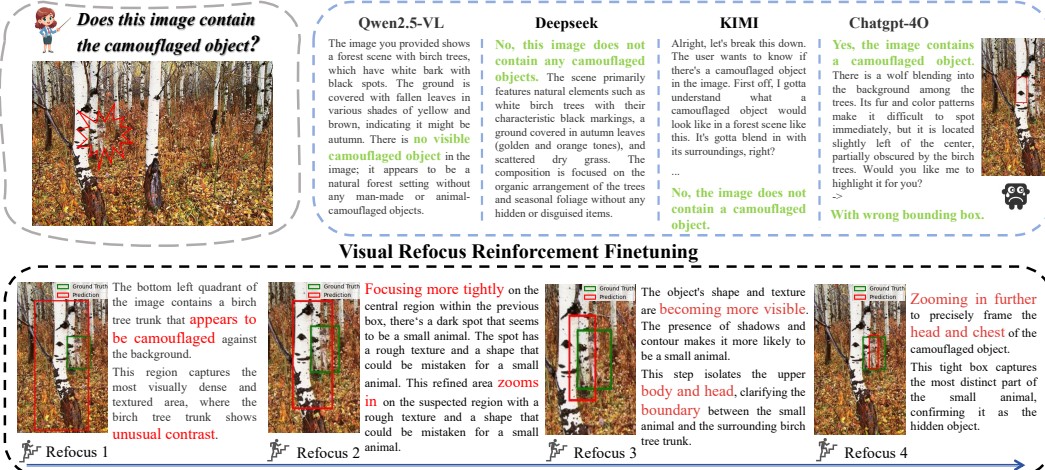

Figure 1: Intriguing discovery of SOTA multi-modal models on limitation: these models struggle to replicate human cognitive processes in leveraging foreground-background similarity relationships for visual analysis. Mimicking human visual camouflaged reasoning perception, our Visual Refocus Reinforcement Fine-Tuning visual system progressively and logically 'refocus' visual concealed content.

Through a comprehensive analysis, as illustrated in Fig. 1, we identify a critical divergence between current multimodal models and human visual cognition in processing challenging camouflaged scenes. Specifically, these models fail to reliably detect objects visually assimilated into the background. Notably, even ChatGPT-4o exhibits hallucinations generating plausible explanations for potential concealment while ultimately failing to localize camouflaged objects. Our findings indicate a critical limitation: their inability to detect visually concealed objects demonstrates a failure to emulate the human cognitive strategy of utilizing foreground-background similarity for visual interpretation.

Motivated by these observations, we are naturally led to explore whether rule-based reinforcement learning approaches can enhance the reasoning capability of Vision-Language Models (VLMs) to mimic the human perception system to iteratively refocus and refine the suspicious zone. We customize a visual 'refocus' curriculum reinforcement learning to learn the visual refocus policy based on the rule-based reward design, which progressively learns a difficulty-hierarchically structured curriculum. Such hierarchical learning mechanism effectively mitigates the directional ambiguity caused by scalar rewards during model exploration. Besides, we also tailor visual in-context refocus reinforcement learning paradigm to capture the context cognitive pattern. This paradigm guides the model in a stepwise manner, enhancing its logical reasoning ability, facilitating the emergence of refocusing capability, and improving both exploration flexibility and controllability. After embedding such mechanisms, we observe the emergence of a "visual refocus" phenomenon, where the localization of concealed objects exhibits a hierarchical attention process that dynamically adjusts and refines prior cognitive representations. The 'visual refocus' phenomenon mainly consists of three observable representation forms: including the 'focus' (*i.e.,* global to local zoom-in), 'rethink' (*i.e.,* local refinement and adjustment of suspicious zone), and 'backtracing' (*i.e.,* from local to global extension retracing). Overall, the main contributions of this paper can be summarized as follows:

- We observe a notable misalignment between existing models and the human system on the perception of camouflaged objects, and propose Visual Refocus Reinforcement Fine-Tuning (VRRF) to align the human camouflaged perception ability.

- We develop a visual 'refocus' curriculum and in-context reinforcement learning paradigm, respectively enabling hierarchical learning of task difficulty and capturing the context-aware cognitive patterns through a rule-based visual refocus reward. Such learning strategy triggers the 'visual refocus' which iteratively and dynamically adjusts and refines prior cognitive knowledge.

- We collect the camouflaged dataset and build the evaluation systems to analyze the camouflaged perception ability. Through comprehensive experiments, we show that our approach to camouflaged object analysis achieves significantly better results than standard supervised fine-tuning (SFT) methods, especially on extremely challenging test set. Furthermore, empirical user studies demonstrate that our visual refocus system outperforms human behavior in camouflaged object perception tasks.

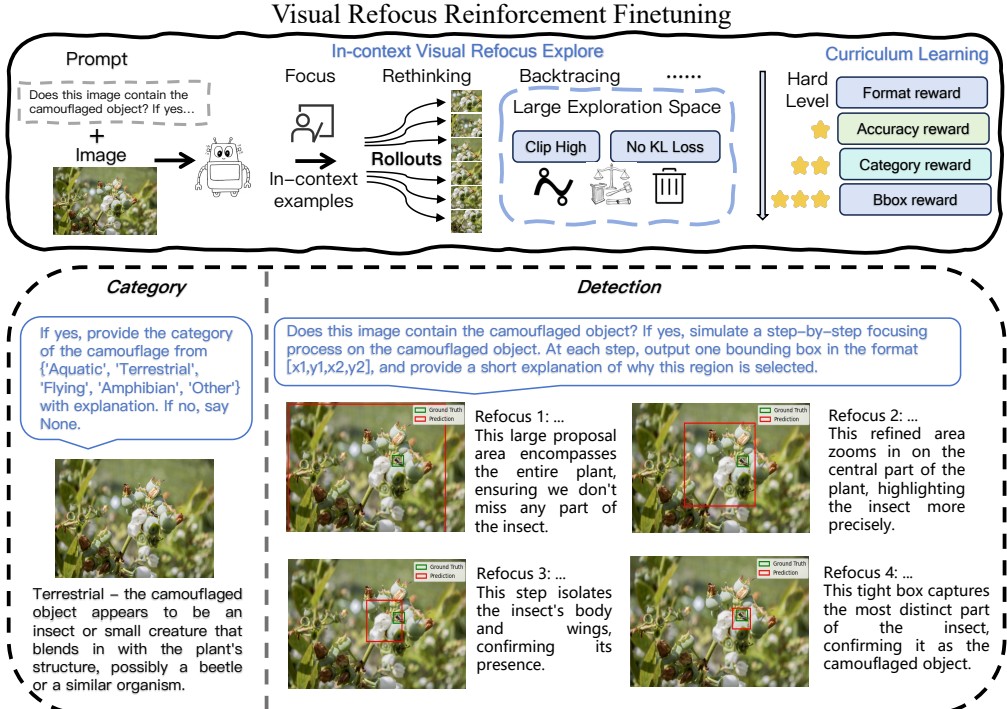

Figure 2: **Overview of Visual Refocus Reinforcement Fine-Tuning**.

## 2 METHODS

In this section, we introduce our approach to stimulating the model's visual refocusing emergence abilities, enabling the model to progressively learn visual reasoning by adaptively switching and refining attention across global and local perspectives, guided by our proposed in-context GRPO framework with refinement.

### 2.1 PRELIMINARY

Group Relative Policy Optimization (GRPO) (Shao et al., 2024a) is a reinforcement learning algorithm designed to fine-tune large language models, particularly in scenarios with sparse or delayed rewards. GRPO simplifies traditional proximal policy optimization by eliminating the need for a separate value function estimator, thus reducing computational overhead. More details of GRPO are provided in the Appendix A.4. While GRPO offers computational advantages, it has shown limited performance in tasks that require multi-step reasoning and fine-grained visual attention, such as camouflaged object detection. Our framework addresses these challenges by introducing a large exploration space with refocus priors, enabling more effective learning in complex visual environments.

**Problem Formulation** Let $x \in \mathcal{X}$ be an image containing a potentially camouflaged object, and let $q \in \mathcal{Q}$ be an accompanying textual prompt (*e.g.*, "Is there a hidden cat?"). Our model maintains an internal chain-of-thought (CoT) state $h_t$ at step $t$, which captures preceding "refocus" actions. We define:

$$s_t = (x, q, h_t), \quad a_t \sim \pi_\theta(a \mid s_t) \tag{1}$$

where $a_t$ is a discrete refocus instruction (*e.g.*, "zoom into region $R$"). After $T$ refocus steps, the model emits a final answer $y$ via a read-out head: $y \sim p_\theta(y \mid s_T)$. Our goal is to learn the policy parameters $\theta$ that maximize the expected utility in identifying the concealed object.

### 2.2 EXPLORATION-AWARE REFOCUS OPTIMIZATION

To enable the model to reason through visually camouflaged scenes, our framework promotes broader and more flexible exploration in both the action and inference spaces. We incorporate two core strategies: in-context reinforcement learning for structured trajectory imitation, and a modified clip-high objective that encourages deviation from prior behavior without regularization penalties.

**In-Context Reinforcement Learning with Trajectory Examples.** Inspired by human learning processes, we design our reinforcement learning policy to facilitate exploration through illustrative examples. Regular GRPO encourages the model to explore autonomously; however, we observed that the performance ceiling is constrained by the model's inherent capabilities. To address this limitation, we integrate explicit in-context demonstrations of visual reasoning steps within each training example, structured specifically to guide and enhance the model's exploration process. We define the example exploration format as in Fig. 3.

```
Does this image contain the camouflaged object?
<refocus instruction>
<format requirement>

# explore
<explore>
==== example i ====
Overview...
Focus (global to local zoom)...
Rethink (local refinement)...
Backtracing (local to global retracing)...
...
Summary.
==== example i+1 ====
...
==== example n ====
...
</explore>
# answers
<bbox>(x=112, y=98, w=64, h=52)</bbox>
<category>Camouflaged Category</category>
<answer>Yes</answer>
```

Figure 3: Prompt example used for in-context reinforcement learning. The `<explore>` block provides a multi-stage visual reasoning trajectory that mimics human perceptual shifts in attention.

Each exploratory example trajectory involves free-form attention adjustments (*e.g.,* zooming, region recognition, refocus) to simulate the iterative process by which humans locate hidden elements. Rather than using structured or fixed exploration steps, we encourage flexible reasoning to enhance the model's capacity for adaptive thinking. This design allows the model to sequentially reason by adaptively attending to previously demonstrated exploratory behaviors, analogous to learning-by-demonstration approaches. The in-context policy $\pi_\theta(a_t|h_{<t}, demo)$ conditions on prior steps $h_{<t}$ and the free-form demonstration. This formulation enables richer generalization without explicitly expanding the model size.

**Clip-High Objective Without KL Penalty.** To encourage broader exploration and escape local optima under the instruction of in-context reinforcement learning, we revise the standard GRPO objective to a *clip-high* variant with removing the KL divergence penalty and instead using a higher clipping ceiling. We define the modified probability ratio as: $r_i(\theta) = \frac{\pi_\theta(o_i|q)}{\pi_{\theta_{\text{old}}}(o_i|q)}$, and the new clip-high loss becomes:

$$L_{\text{GRPO}}^{\text{clip-high}}(\theta) = \mathbb{E}_{q\sim D,\, o_i\sim\pi_{\theta_{\text{old}}}(\cdot|q)} \Big[ \min\Big(r_i(\theta)\hat{A}_i,\ \text{clip}(r_i(\theta), 1-\epsilon, 1+\delta)\hat{A}_i\Big) \Big], \quad (2)$$

where $\delta > \epsilon$ allows for a looser upper bound on policy shifts while still preventing collapse. By removing the KL term and expanding the upper clipping range, this formulation encourages the model to explore less likely (but potentially more optimal) paths and reduces the over-penalization of novel or rare reasoning trajectories. We find that this modification leads to improved localization of highly camouflaged content and less reliance on prior policy or SFT biases.

## 2.3 CURRICULUM REINFORCEMENT LEARNING FOR PROGRESSIVE REWARD ACQUISITION

In GRPO, the final reward is computed as the sum of multiple scalar rewards. This aggregation obscures detailed feedback on individual reward components, making it difficult to precisely discern which rewards are increasing or decreasing. Consequently, the model faces challenges in effectively

learning the format or improving accuracy. To further guide the model in mastering progressively harder aspects of camouflaged perception, we introduce a curriculum-style reinforcement learning schedule that incrementally augments the reward signal. Concretely, we define three stages—*format & accuracy*, *category*, and *localization IoU*—and successively incorporate them into the group relative policy optimization (GRPO) objective.

**Stage 1: Format and Accuracy Rewards.** In the initial phase, we focus the model on producing well-formed outputs and achieving basic recognition correctness. For each candidate output $o_i$, we compute:

$$R_i^{(1)} = \lambda_{\text{fmt}} R^{\text{fmt}}(o_i) + \lambda_{\text{acc}} R^{\text{acc}}(o_i), \tag{3}$$

where $R^{\text{fmt}}(o_i) \in [0,1]$ penalizes malformed XML tags or missing tokens in the `<bbox>`, `<category>`, and `<answer>` fields, $R^{\text{acc}}(o_i) \in \{0,1\}$ grants 1 point if the predicted $y$ matches ground-truth presence, and $\lambda_{\text{fmt}}, \lambda_{\text{acc}}$ are scaling coefficients. We then compute the group-relative advantage $\hat{A}_i^{(1)}$ using these $R_i^{(1)}$ values exactly as in Section 2.1, and optimize the clip-high objective.

**Stage 2: Adding Category Reward.** Once the model reliably produces syntactically valid outputs and answers presence questions with the predefined format, we introduce semantic category correctness. Denote:

$$R^{\text{cat}}(o_i) = \begin{cases} 1, & \text{if predicted category matches ground truth,} \\ 0, & \text{otherwise.} \end{cases} \tag{4}$$

The cumulative reward becomes:

$$R_i^{(2)} = \lambda_{\text{fmt}} R^{\text{fmt}}(o_i) + \lambda_{\text{acc}} R^{\text{acc}}(o_i) + \lambda_{\text{cat}} R^{\text{cat}}(o_i), \tag{5}$$

and we form the corresponding relative advantages $\hat{A}_i^{(2)}$ from stage-2.

**Stage 3: Incorporating IoU Refinement.** Finally, to refine localization quality, we append an Intersection-over-Union (IoU) reward:

$$R^{\text{iou}}(o_i) = \frac{\text{area}(\text{pred}_i \cap \text{gt})}{\text{area}(\text{pred}_i \cup \text{gt})}, \tag{6}$$

and define the full-stage reward:

$$R_i^{(3)} = \lambda_{\text{fmt}} R^{\text{fmt}}(o_i) + \lambda_{\text{acc}} R^{\text{acc}}(o_i) + \lambda_{\text{cat}} R^{\text{cat}}(o_i) + \lambda_{\text{iou}} R^{\text{iou}}(o_i). \tag{7}$$

With $\hat{A}_i^{(3)}$ computed in the usual way, our final curriculum objective is:

$$L^{(3)}(\theta) = \mathbb{E}_{q,o_i} \left[ \min \left( r_i(\theta) \hat{A}_i^{(3)}, \text{clip}(r_i(\theta), 1 - \epsilon, 1 + \delta) \hat{A}_i^{(3)} \right) \right]. \tag{8}$$

We transition from one stage to the next when the reward ceases to increase, typically after approximately 2–6 epochs. In practice, we set $\lambda_{\text{fmt}} = \lambda_{\text{acc}} = \lambda_{\text{cat}} = \lambda_{\text{iou}} = 1$ without tuning trade-off parameters. This progressive pipeline enables the model to first master output structure and basic presence awareness, then semantic classification, and ultimately fine-grained localization. By structuring reward signals as a curriculum, our framework guides the model through increasingly complex visual reasoning tasks—mirroring the progressive 'refocusing' characteristic of human perception—while preserving the computational efficiency of GRPO's relative advantage formulation.

## 3 EXPERIMENTS

### 3.1 EXPERIMENTAL SETUP

**Implementation Details.** We use Qwen-2.5-VL-7B (Bai et al., 2025a) as the base model. The default GRPO settings are adopted, with the number of generations N=4, temperature set to 1, and the clip high value fixed at 0.28. We train the model using the AdamW, starting with a learning rate of 1e-6, which is linearly decayed over the course of training. The model is trained with batch size of 8. Training is completed within approximately one day using 8 NVIDIA H20 GPUs.

Table 1: Quantitative evaluation results on Easy-Concealed Object test set, decomposed into camouflaged object classification and detection tasks, comparing GPT-4.1 (Achiam et al., 2023), Qwen2.5-VL (7B/72B) (Bai et al., 2025b), InternVL3 (8B) (Zhu et al., 2025) and R1-V (Chen et al., 2025). The best results are highlighted in **bold** and the second-best is marked in underline.

| Easy Concealed Object Set | Existence Binary Acc ↑ | Concealed Category Classification | | | |
| --- | --- | --- | --- | --- | --- |
| | | Category Acc ↑ | Precision ↑ | Recall ↑ | F1 ↑ |
| GPT4.1 | 0.99 | 0.82 | 0.87 | 0.82 | 0.81 |
| InternVL3-8B | 0.94 | 0.54 | 0.50 | 0.49 | 0.44 |
| Qwen2.5-vl-7B | 0.84 | 0.56 | 0.53 | 0.56 | 0.52 |
| Qwen2.5-vl-72B | 0.95 | 0.65 | 0.80 | 0.65 | 0.63 |
| R1-V | 0.97 | 0.63 | 0.50 | 0.63 | 0.52 |
| SFT | 0.97 | 0.84 | 0.89 | 0.84 | 0.86 |
| VRRF | **0.99** | **0.89** | **0.90** | **0.89** | **0.90** |

| Easy Concealed Object Set | F1@0.5 ↑ | Concealed Detection | | | | | |
| --- | --- | --- | --- | --- | --- | --- | --- |
| | | Preicision@0.5 ↑ | Recall@0.5 ↑ | mIOU ↑ | IoU ≥ 0.3(%) ↑ | IoU ≥ 0.7(%) ↑ | Mean Center Distance(px) ↓ |
| GPT4.1 | 0.37 | 0.37 | 0.37 | 0.41 | 66.67 | 8.82 | 141.17 |
| InternVL3-8B | 0.44 | 0.51 | 0.38 | 0.33 | 51.96 | 12.75 | 142.52 |
| Qwen2.5-vl-7B | 0.26 | 0.34 | 0.21 | 0.24 | 32.35 | 13.73 | 93.90 |
| Qwen2.5-vl-72B | 0.49 | 0.51 | 0.48 | 0.47 | 69.61 | 25.49 | 57.72 |
| R1-V | 0.54 | 0.56 | 0.52 | 0.50 | 69.61 | 35.29 | 162.75 |
| SFT | 0.60 | 0.62 | 0.59 | 0.53 | 81.37 | 29.41 | 110.96 |
| VRRF | **0.88** | **0.88** | **0.88** | **0.75** | **95.10** | **74.51** | **35.98** |

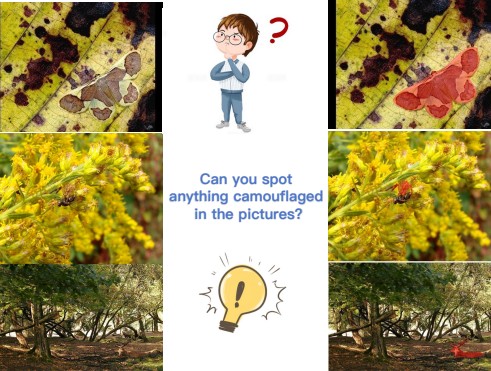

Figure 4: Examples from hard-concealed set. Can you find them? Best viewed in color and zoomed-in.

**Datasets and Evaluation Metrics.** We evaluate our proposed VRRF on four public benchmark datasets for Camouflaged Object Detection (COD): COD10K (Fan et al., 2020), NC4K (Lv et al., 2021), CAMO (Le et al., 2019) and CHAMELEON (Skurowski et al., 2018), covering diverse challenging camouflage scenarios. To assess model performance under varying levels of difficulty, we construct two subsets: easy and hard concealed object testsets in Fig. 4. The hard testset is extremely challenging and is manually selected from the full datasets. The easy set is sampled images from the remaining data. In total, our experimental data includes 14,017 training samples (comprising 9,083 camouflaged and 4,934 noncamouflaged). For the category classification task, we adopt the five super-categories defined in the COD10K dataset: Aquatic, Terrestrial, Flying, Amphibian, and Other, and instruct the model to choose from these classes for consistent labeling. For the detection task, bounding boxes are derived from the provided segmentation masks.

## 3.2 CAMOUFLAGED OBJECT PERCEPTION

**Camouflaged Object Perception and Category Classification.** As shown in Tab. 2, on the challenging hard dataset, even human struggle to recognize these difficult camouflaged cases within seconds. Our method outperforms Supervised Fine-Tuning (SFT) on the category classification and concealed object perception primarily by modeling the reasoning process rather than supervising only the final output. Our Visual Refocus Policy training paradigm guides the model to progressively localize objects through intermediate bounding box predictions (*e.g.,* focusing on discriminative parts like animal heads/chests/wings). This process-oriented approach, combining stepwise visual grounding with classification, significantly improves final classification and perception accuracy.

**Camouflaged Object Detection.** On the hard concealed test set (Tab. 2), VRRF achieves a markedly larger performance gain than on the easy set (Tab. 1), suggesting that more challenging tasks require stronger reasoning ability and gain substantial improvements from a well-equipped reasoning model. Notably, our system even surpasses human performance in the hard cases, as confirmed by user studies where humans struggle to accurately localize and categorize camouflaged objects within limited time. The superior performance is primarily attributed to our novel mechanism - an integrated framework combining reasoning, joint training, and refocusing capability, which enables progressive localization of camouflaged objects in complex environments through iterative reasoning.

## 3.3 VISUAL REFOCUS MECHANISM

**Visual Refocus Analysis.** Our visual refocus mainly demonstrates three refocus paradigms including the 'Focus' (*i.e.,* global to local zoom-in), 'Rethink' (*i.e.,* local refinement) and 'Backtracing' (*i.e.,*

Table 2: Quantitative evaluation results on Hard-Concealed Object test set, decomposed into camouflaged object classification and detection tasks, comparing GPT-4.1 (Achiam et al., 2023), Qwen2.5-VL (7B/72B) (Bai et al., 2025b), InternVL3 (8B) (Zhu et al., 2025), and R1-V (Chen et al., 2025). The best results are highlighted in **bold** and the second-best is marked in underline.

| Hard Concealed Object Set | Existence Binary Acc ↑ | Concealed Category Classification | | | |
|---|---|---|---|---|---|
| | | Category Acc ↑ | Precision ↑ | Recall ↑ | F1 ↑ |
| Human perception | 0.65 | 0.46 | 0.78 | 0.46 | 0.58 |
| GPT4.1 | 0.91 | 0.78 | 0.85 | 0.78 | 0.79 |
| InternVL3-8B | 0.79 | 0.45 | 0.50 | 0.45 | 0.44 |
| Qwen2.5-vl-7B | 0.42 | 0.28 | 0.73 | 0.28 | 0.38 |
| Qwen2.5-vl-72B | 0.50 | 0.44 | 0.89 | 0.44 | 0.58 |
| R1-V | 0.79 | 0.59 | 0.74 | 0.59 | 0.62 |
| SFT | 0.83 | 0.70 | 0.86 | 0.70 | 0.75 |
| VRRF | **0.92** | **0.80** | 0.88 | **0.80** | **0.82** |

| Hard Concealed Object Set | Concealed Detection | | | | | | |
|---|---|---|---|---|---|---|---|
| | F1@0.5 ↑ | Preicision@0.5 ↑ | Recall@0.5 ↑ | mIOU ↑ | IoU ≥ 0.3(%) ↑ | IoU ≥ 0.7(%) ↑ | Mean Center Distance(px) ↓ |
| Human perception | 0.51 | **0.65** | 0.42 | 0.38 | 49.04 | 32.69 | 69.82 |
| GPT4.1 | 0.21 | 0.21 | 0.20 | 0.24 | 40.38 | 3.85 | 143.88 |
| InternVL3-8B | 0.16 | 0.21 | 0.13 | 0.15 | 23.08 | 4.81 | 188.48 |
| Qwen2.5-vl-7B | 0.16 | 0.33 | 0.11 | 0.12 | 15.38 | 7.69 | 83.80 |
| Qwen2.5-vl-72B | 0.34 | 0.52 | 0.25 | 0.25 | 33.65 | 15.38 | **55.16** |
| R1-V | 0.32 | 0.37 | 0.29 | 0.29 | 40.38 | 15.38 | 135.17 |
| SFT | 0.42 | 0.48 | 0.38 | 0.37 | 53.85 | 20.19 | 89.98 |
| VRRF | **0.54** | 0.58 | **0.50** | **0.47** | 60.58 | 38.46 | 61.11 |

Table 3: Evaluating the generalization ability of our VFFR on detection, hallucination and general benchmark.

| Method | Refcoco | | | Refcoco+ | | | Refcocog | | POPE | | | HRbench-4K | | | HRbench-8K | | |
|---|---|---|---|---|---|---|---|---|---|---|---|---|---|---|---|---|---|
| | val | testA | testB | val | testA | testB | val | test | Adv. | Rand. | Pop. | FSP | FCP | Overall | FSP | FCP | Overall |
| Qwen2.5-VL | 0.88 | 0.92 | 0.85 | 0.84 | 0.89 | 0.76 | 0.87 | 0.87 | 0.81 | 0.82 | 0.81 | **0.82** | 0.55 | 0.68 | **0.75** | 0.52 | 0.63 |
| VRRF | **0.90** | **0.93** | **0.86** | 0.84 | 0.89 | **0.78** | 0.87 | **0.88** | **0.84** | **0.86** | **0.85** | 0.80 | **0.59** | **0.70** | 0.71 | **0.55** | 0.63 |

Table 4: Evaluation results of detection model *vs.* our VRRF on easy camouflaged objects detection (Easy COD) and hard camouflaged objects detection (Hard COD) dataset.

| Dataset | Methods | mIOU ↑ | IoU ≥ 0.3(%) | IoU ≥ 0.7(%) | Precision@0.5 ↑ | Recall@0.5 ↑ | F1@0.5 ↑ |
|---|---|---|---|---|---|---|---|
| Easy COD | Grounding Dino | 0.41 | 46.46 | 45.92 | **0.94** | 0.46 | 0.62 |
| | YOLOV11 | **0.79** | 92.16 | **80.39** | 0.60 | **0.91** | 0.72 |
| | VRRF | 0.75 | **95.10** | 74.51 | 0.88 | 0.88 | **0.88** |
| Hard COD | Grounding Dino | 0.25 | 30.77 | 28.41 | **0.66** | 0.30 | 0.41 |
| | YOLOV11 | 0.40 | 52.68 | 24.11 | 0.25 | 0.41 | 0.31 |
| | VRRF | **0.47** | **60.58** | **38.46** | 0.58 | **0.50** | **0.54** |

local to global focus). During stepwise localization, the initial bounding box typically adopts a coarse-grained scope to ensure comprehensive coverage of potential clues. As a 'Focus' mechanism, subsequent steps progressively focus with the reasonable tokens (*i.e.,,* zoom in) to capture finer details (*e.g.,* head/chest) until converging to the final precise box. Notably, our framework supports 'Backtracing' refinement paradigms. For instance, as shown in Fig. 5 - 4-th row, the model may first attend to discriminative local features of the cicada at intermediate steps, such as its head and wings, which stand out due to their distinct shape and color, and then zoom out to capture the whole object.

**Extending to Multiple Object Perception.** Fig. 6 demonstrates our model's ability to generalize to multiple-objects. During reasoning, the model is able to detect multiple objects within the same scene and simultaneously refine each predicted bounding box. This demonstrates the 'Rethink' capability, allowing iterative optimization of multiple predictions while maintaining accurate localization.

**Generalization ability analysis of VFFR on general, hallucination and detection benchmark.** In general high-relevant visual grounding benchmarks (*e.g.,* Refcoco/+/g ) as shown in Tab. 3, our VRRF performs better than the Qwen2.5-VL baseline, while our VRRF also maintains its comparable general ability on HRBench. It indicates that the training paradigm based on the foreground-background similarity principle also facilitates the improvement of general object detection. In the hallucination benchmark (*e.g.,* POPE), our 'refocus' visual mechanism develops a correction ability to rethink its behaviors and alleviate its visual hallucination to some extent.

**Comparisons with other detection models.** In addition, we further compare our method with state-of-the-art object detection models, including YOLOv11 (Khanam & Hussain, 2024) and Grounding DINO (Liu et al., 2024b). YOLO is retrained on the same camouflage dataset, while Grounding DINO is evaluated under identical settings. The three models display different tendencies in Table 4:

**Does this image contain the camouflaged object? If yes, simulate a step-by-step focusing process on the camouflaged object. At each step, output one bounding box in the format [x1,y1,x2,y2], and provide a short explanation of why this region is selected.**

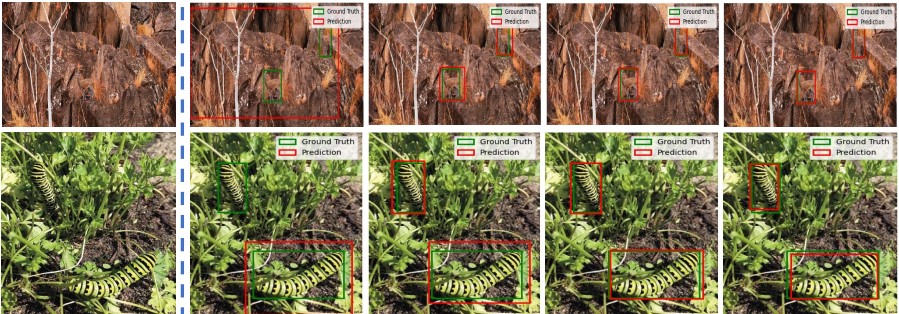

Figure 5: **Illustration of 'Visual Refocus' representation pattern**. The first three rows show 'focus' in the form of global to local zoom-in, 4-th row denotes 'backtracing' from local to global extension retracing after perceiving the discriminate head and wing part.

Figure 6: **Illustration of our VRRF's generalization to multiple object perception**. Our VRRF inherits 'rethink' capability to refine and adjust the box to detect multiple objects.

Table 5: Qualitative results of the ablation study on the Hard-Concealed Object test set.

| Hard Concealed Object Set | Concealed Object Existence Binary Acc | Classification Category Acc | F1 | Detection mIOU | IoU $\geq$ 0.5(%) | F1@0.5 |
|---|---|---|---|---|---|---|
| Qwen2.5-vl (Bai et al., 2025b) | 0.423 | 0.28 | 0.38 | 0.120 | 10.58 | 0.16 |
| + format & acc reward | 0.789 | - | - | - | - | - |
| + category reward | 0.837 | 0.73 | 0.79 | - | - | - |
| + bbox reward | 0.856 | 0.74 | 0.79 | 0.420 | 42.31 | 0.47 |
| + in-context learning | **0.923** | **0.80** | **0.82** | **0.473** | **54.81** | **0.54** |
| All-at-once (No Curriculum) | 0.788 | 0.65 | 0.72 | 0.374 | 41.35 | 0.47 |

YOLO favors higher recall but suffers from low precision due to more false positives (mIoU, defined in Appendix A.5, considers only the best-matching box per ground-truth, partly explaining YOLO's higher scores in easy COD), whereas Grounding DINO is conservative, predicting fewer boxes with high precision but low recall. Our proposed VRRF achieves a better balance between these two extremes, leading to the best overall performance. Notably, the advantage of our model becomes more evident on the more challenging test sets due to the 'refocus' mechanism.

## 3.4 ABLATION STUDIES

**Ablation of Each Component.** As shown in Tab. 5, we conduct comprehensive ablations progressively adding the refocus policy including the format, accuracy, category and bbox reward mechanism,

Table 6: Qualitative results of the clip-high objective ablation study on the Hard-Concealed Object test set. The best results are highlighted in **bold**.

| Hard Concealed Object Set | Concealed Object Existence Binary Acc | Classification | | Detection | | |
|---|---|---|---|---|---|---|
| | | Category Acc | F1 | mIOU | IoU $\geq 0.5$(%) | F1@0.5 |
| VRRF (w/o kl and clip_upper_bound=0.28) | **0.923** | **0.80** | **0.82** | **0.473** | **54.81** | **0.54** |
| with KL_loss | 0.827 | 0.78 | 0.79 | 0.470 | 52.88 | 0.52 |
| low_clip_upper_bound=0.2 | 0.846 | 0.72 | 0.77 | 0.461 | 50.96 | 0.52 |

and the in-context learning reinforcement learning. We observe that the addition of each reward gradually boosts both the classification accuracy and the detection performance. Such joint training framework enhances the model's reasoning capability through iterative refinement. In-context learning further guides the model on how to think by teaching it to dynamically refocus attention to detect camouflaged objects, verify intermediate results, and progressively refine predictions (*e.g.,* through refocus adjustments based on prior outputs). In contrast, adding all components simultaneously results in inferior performance, likely due to the complexity of jointly optimizing the camouflaged object perception tasks, and therefore highlighting the benefit of our progressive learning.

**Ablation of Training Strategies.** We present additional ablation studies evaluating the impact of the KL penalty and different upper clipping bounds in Tab. 6. Clip-High objective without KL penalty leads to better classification and detection performance by enabling the model to explore low-probability reasoning paths that may yield more optimal results.

**Analysis of Runtime and Accuracy in Different Refocusing Steps.** With the increase in refocusing steps, the inference time also grows gradually. The mIOU peak lies at the 4-th refocusing step. In our experiment, we adopt the four refocusing steps as our default setting.

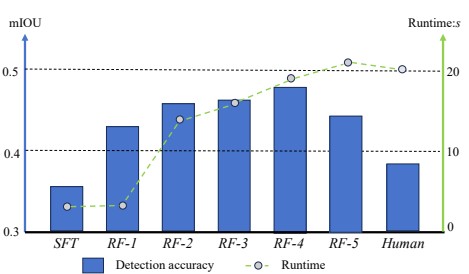

Figure 7: Impact of Refocusing Steps on Inference Time and Performance.

**User Study and Explanation of VRRF surpass human perception.** Thirty well-educated participants with healthy visual systems are trained by the labeling process (*e.g.,* the teaching of 'labelme' tool operation; and the demonstration of camouflaged object detection). We testify that twenty seconds are sufficient to traverse the pixel-wise information twice, so we adopt this setting to avoid excessive visual fatigue. Participants are asked to detect the camouflaged object and classify the specific class. Under this setting, the user study outperforms the SFT baseline but remains below VRRF. The potential reason why our model mimics human principles of discerning foreground-background discrepancies and then excels the human perception can be summarized as: *1).* Human attention is inherently limited and often overlooks subtle details — such as minor textures or faint color gradients — that may be critical in extreme camouflage (*e.g.*, certain insects or military patterns). In contrast, the model detects such fine-grained signals consistently and sensitively through computational analysis of pixel-level data, surpassing human reliance on contextual reasoning and educated guesses. *2).* The VRRF model operates without human limitations such as fatigue, emotional variability, or fluctuating concentration levels due to external factors. It applies relentless, uniform attention to every sample, enabling large-scale, highly consistent processing beyond human capacity.

## 4 CONCLUSION

In this paper, we observe that current multimodal models fail to detect concealed objects, lacking the human cognitive ability to analyze foreground-background similarity. We address the critical gap between multimodal models and human cognition in detecting camouflaged objects by introducing a visual refocus reinforcement framework. By emulating human-like hierarchical reasoning to progressively shift attention, our method enables models to dynamically refine and refocus their predictions through stepwise analysis. Extensive experiments demonstrate that this approach not only outperforms SFT baselines in classification and detection tasks but also exhibits emergent human-aligned refocusing behaviors, characterized by multi-token reasoning. Our work provides a pathway toward more cognitively inspired multimodal systems.

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

## A APPENDIX

### A.1 REPRODUCIBILITY STATEMENT

We have already elaborated on all the models or algorithms proposed, experimental configurations, and benchmarks used in the experiments in the main body or appendix of this paper. Furthermore, we declare that the entire code used in this work will be released after acceptance.

### A.2 THE USE OF LARGE LANGUAGE MODELS

We use large language models solely for polishing our writing, and we have conducted a careful check, taking full responsibility for all content in this work.

### A.3 RELATED WORK

**Vision-Language Models.** Vision-Language Models (VLMs) have witnessed remarkable development since the emergence of large language models (LLMs). Pioneering works such as Flamingo Alayrac et al. (2022) lay the foundation for VLMs, demonstrating their potential in few-shot learning tasks. Subsequently, LLaVA Liu et al. (2024a) employs GPT-4 Achiam et al. (2023) to generate training data, achieving promising results in visual dialogue and reasoning, which inspired a series of studies focusing on visual instruction data, like InstructBLIP Dai et al. (2023). Besides, some studies aim to enhance visual reasoning capabilities through high-resolution inputs Shi et al. (2025), improved spatial understanding Chen et al. (2024a), and the modeling of human visual illusions Zhang et al. (2023). To address the limitation of constrained image input resolution in early VLMs, mechanisms like AnyRes Chen et al. (2024b;d) are introduced, enabling flexible handling of images with different resolutions and aspect ratios and enhancing the models' perceptual and reasoning capabilities. Currently, popular open-source VLM series include LLaVA Li et al. (2024); Liu et al. (2024a), QwenVL Bai et al. (2025b); Wang et al. (2024), and InternVL Chen et al. (2024c;d). Building upon existing VLMs and inspired by the observed discrepancy between human and VLMs, we explore this interesting phenomenon and develop a visual refocus system to better bridge and surpass the human camouflaged perception ability.

**Reinforcement Learning in Vision-Language Models.** The application of reinforcement learning (RL) in Vision-Language Models has become an active research area. DeepSeek R1 Guo et al. (2025b) demonstrated that simple rule-based rewards can significantly enhance the reasoning capabilities of LLMs, inspiring researchers to explore the extension of similar RL methodologies to VLMs. Works like R1-OneVision Yang et al. (2025b) proposed a cross-modal reasoning pipeline to improve VLM reasoning, while R1-V Chen et al. (2025) introduced the GRPO method Shao et al. (2024b) into

VLM training for object-counting tasks. VisualThinker-R1-Zero Zhou et al. (2025b) showed that applying RL to base VLMs can lead to substantial performance improvements and trigger the "visual aha moment". Most of these previous studies on RL in VLMs target common visual understanding tasks or multimodal mathematics tasks. In contrast, our research focuses on the better visual reasoning alignment between humans and VLMs especially on the camouflaged object perception with unique challenges due to the nature of camouflaged objects blending into the background. Our visual refocus-based RL approach is designed to guide the model to perform a human-like chained thinking and visual focusing process, enabling dynamic adjustment and refinement of prior cognitive representations.

**Camouflaged Object Perception.** Camouflaged object perception, a bio-inspired research field, focuses on detecting concealed objects or animals that visually blend into their surroundings Fan et al. (2021); Hu et al. (2023); Tang et al. (2024). Biological and psychological studies Cuthill (2019); Stevens & Merilaita (2009) demonstrate that camouflage serves as both a survival mechanism for prey species to evade predators and a perceptual challenge for human vision systems, which are particularly sensitive to edge-related color and illumination cues. Investigating camouflage phenomena offers valuable insights into the fundamental mechanisms of human visual perception. Our method introduces multi-modal large models with RL and visual refocus technology, endowing the model with human-like high-level thinking and visual focusing abilities. This allows the model to not only analyze the visual features of the image but also conduct reasoning based on semantic information.

## A.4    DETAILS OF GROUP RELATIVE POLICY OPTIMIZATION (GRPO)

In GRPO, for a given input (e.g., an image-question pair), the policy generates a group of $G$ outputs $\{o_1, o_2, \ldots, o_G\}$. Each output $o_i$ receives a scalar reward $R_i$, possibly from a reward model or human feedback. The average reward for the group is computed as: $\bar{R} = \frac{1}{G} \sum_{i=1}^{G} R_i$. The relative advantage for each output is then: $\hat{A}_i = (R_i - \bar{R})/std(R)$. This approach focuses on the relative performance of each output within the group, promoting outputs that perform better than average.

The GRPO objective function is:

$$L_{\text{GRPO}}^{\text{clip}}(\theta) = \mathbb{E}_{q \sim D,\, o_i \sim \pi_{\theta_{\text{old}}}(\cdot|q)} \Big[ \min \Big( r_i(\theta)\hat{A}_i,\ \text{clip}(r_i(\theta), 1 - \epsilon, 1 + \epsilon)\hat{A}_i \Big)$$
$$- \beta \cdot D_{\text{KL}}\big[\pi_\theta(\cdot \mid q) \,\|\, \pi_{\text{ref}}(\cdot \mid q)\big] \Big], \tag{9}$$

where $\beta$ controls the strength of the regularisation term, $q$ is the sampled question and $D_{\text{KL}}$ is the Kullback-Leibler divergence between the current policy and a reference policy.

## A.5    EVALUATION METRICS

We summarize here the evaluation metrics used to assess classification and detection performance in our experiments. To evaluate the classification task, we use the following metrics: Category Accuracy, weighted precision, weighted recall, and weighted F1 score. These provide complementary perspectives on the classification performance across all categories. For the detection task, we report Precision@0.5, Recall@0.5, and F1@0.5, as well as mIoU and IoU@0.3/0.5/0.7. For Precision, Recall and F1, evaluations are computed under the standard detection protocol, where predictions with IoU > 0.5 are regarded as true positives, thereby capturing the trade-off between over- and under-prediction. In contrast, mIoU and IoU@X follow a best-match strategy, where only the predicted box with the highest overlap is assigned to each ground-truth instance, independent of the total number of predictions. This ensures that these metrics measure the overlap quality between predicted and ground truth bounding boxes, capturing both coarse and fine-grained localization performance.

## A.6    LIMITATIONS AND BROADER IMPACTS

Our proposed visual refocus reinforcement learning framework effectively enhances reasoning and cognitive capabilities by prompting MLLMs to refocus attention before answering. While our method can positively impact applications like image analysis through improved visual perception, it also risks misuse in privacy-invasive visual systems, highlighting ethical considerations for responsible usage. Our visual refocus mechanism is primarily designed for specific camouflaged perception scenarios, which are commonly encountered in fields such as computer vision (*e.g.,* search and rescue operations or rare species discovery), medical imaging (*e.g.,* polyp detection and lung infection

identification), and agriculture (*e.g.,* locust detection for invasion prevention). A more general refocus mechanism will be explored in future work.

## A.7 "VISUAL REFOCUS" REPRESENTATION PATTERN

We provide more detailed text and image illustrations of "Visual Refocus" representation pattern, as shown in Fig. 8

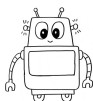

**Does this image contain the camouflaged object? If yes, simulate a step-by-step focusing process on the camouflaged object. At each step, output one bounding box in the format [x1,y1,x2,y2], and provide a short explanation of why this region is selected.**

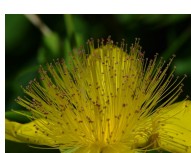

The initial large box encompasses the **entire flower**, including the stamens and the central area where the object is located. This large box ensures ....

The **final**, precise bounding box identifies the object as a small insect or spider, which blends into the flower's coloration. This tight box confirms the location of the camouflaged object, which is a small insect or spider.

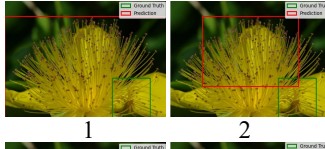

1    2

4    3

**Refining** the **focus on** the central area where the object is located, we zoom in on the **stamen region**. This refined box narrows down the focus to the stamen region, where the object is situated

Further **refining** the focus, we **zoom in** on the specific area around the object, which appears to be a small insect or spider. This step aims to isolate the object more precisely, focusing on the area where the object is located

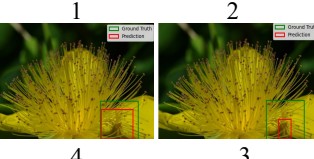

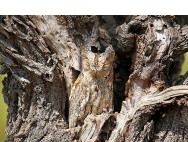

... The bird blends in with the bark, making it challenging to spot initially. This large proposal area encompasses the entire tree trunk, ensuring we don't miss any part of the bird.

**Zooming in** further to precisely frame the bird's **entire body**, including its **wings and tail**, which are also camouflaged. This tight box captures the bird's entire body, confirming it as the camouflaged object

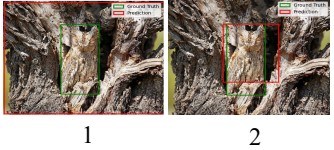

1    2

4    3

Focusing on the central region of the trunk, the bird's head and upper body become more visible due to the texture and color of the bark. This refined area zooms in on the bird's head and upper body, highlighting the bird's integration with the bark.

The bird's eyes and beak are now more clearly defined, showing the bird's facial features and the contrast with the bark. This step isolates the bird's head and upper body, confirming the bird's presence and its camouflage against the bark

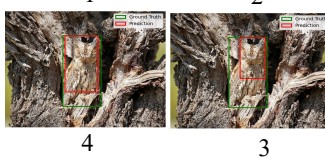

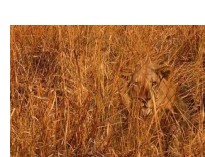

... A large area needs to be considered for potential camouflage. This large box encompasses the entire image, including the lion partially hidden in the grass. It helps us understand the context and the extent of the camouflage.

The final, precise bounding box captures the lion's face and upper body, confirming the camouflage. This tight box confirms the lion's face and upper body, showing how well it blends into the grassy environment.

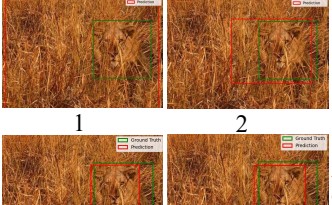

1    2

4    3

The lion's entire body becomes more visible as the camera zooms in on the entire image. This step confirms the lion's location and its level of camouflage. The lion's fur blends seamlessly with the dry grass, making it difficult to distinguish at first glance

Further refining the focus on the lion's face, we can see the details more clearly. This step zooms in even closer on the lion's face, showing the details of its eyes and fur pattern.

Figure 8: Detailed text and image illustrations of 'Visual Refocus' representation pattern.

## A.8 VISUAL REASONING ON CONCEALED CATEGORY CLASSIFICATION

We offer more visual reasoning results on concealed category classification task between the SFT and our VRRF, as shown in Fig. 9.

## A.9 USER STUDIES ON 'LABELME'

We also conduct user studies labeling on the 'Labelme' tool to classify the concealed category and detect the bounding-box in Fig. 10.

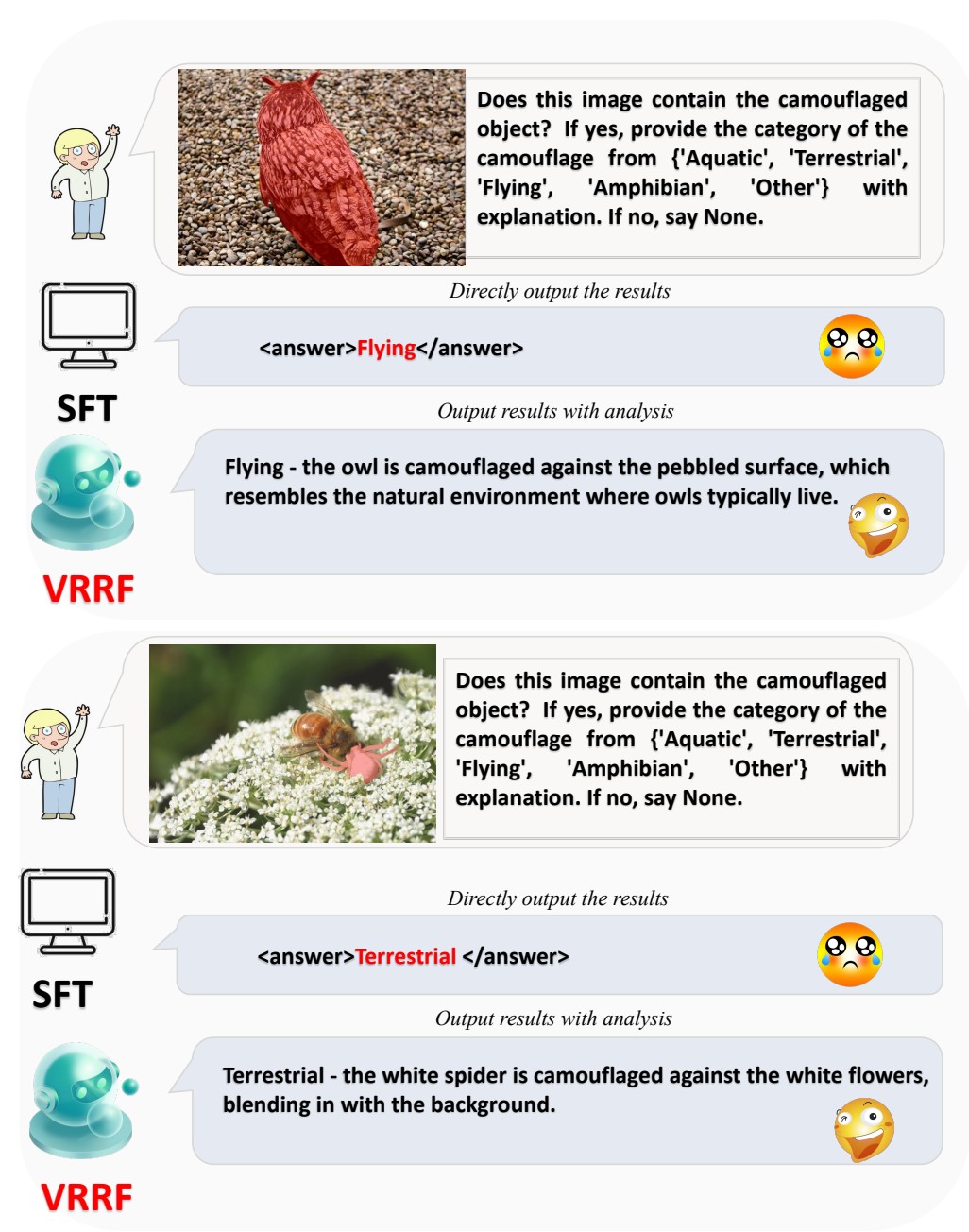

Figure 9: Visual reasoning results on concealed category classification task between the SFT and our VRRF.

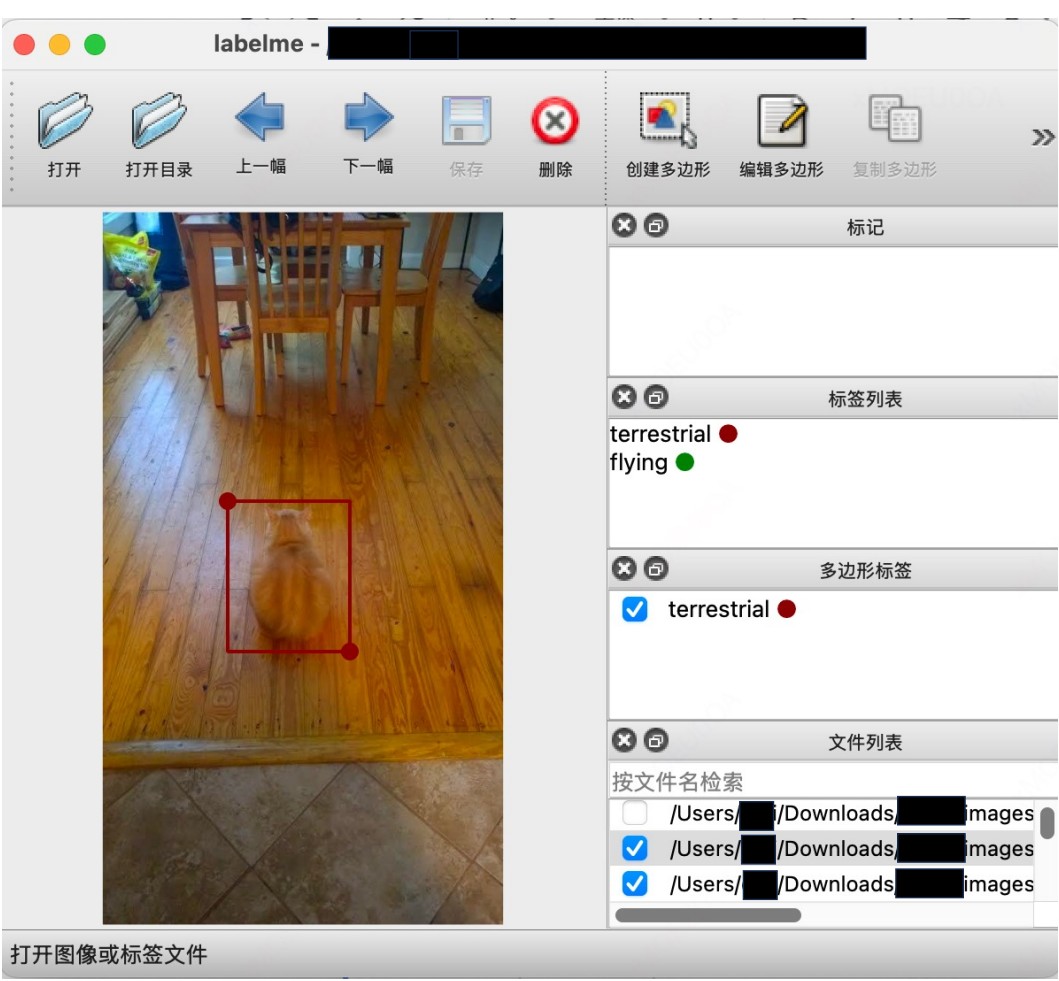

Figure 10: User studies: Human annotations on 'Labelme' tool.

