# OpenReview forum: "Align Human Camouflaged Perception: Visual Refocus Reinforcement Fine-Tuning"
_ICLR.cc/2026/Conference — ICLR 2026 Conference Withdrawn Submission_

### Official Review · Reviewer_GTD9 · 2025-10-29

**Soundness:** 2
**Presentation:** 1
**Contribution:** 2
**Rating:** 4
**Confidence:** 4

**Summary:**

This paper proposes a novel framework (VRRF) for enhancing the performance of VLMs on identifying camouflage animals from the Concealed Object dataset. The authors make use of a modified (Group Relative Policy Optimization) GRPO with a curriculum with progressive rewards during training. The results are benchmarked against other models and human subjects on identification and classification of the camouflaged objects. A noticeable improvement is seen from the application of the VRRF RL framework towards Qwen2.5. Further ablation studies demonstrate the efficacy and importance of the curriculum as well as various modifications of GRPO.

**Strengths:**

The paper is well motivated and the results are presented comprehensively.

The curriculum was interesting and a novel application to camouflage (as far as I know), with prompt engineering adapted to the problem.

The progressive reward acquisition is also demonstrated thoroughly with a set of ablation studies.

There are concrete examples throughout the paper that make the framework generally understandable ovearll (though there are points of confusion in specific areas)

The analysis of effect of refocus steps on increasing inference time is a useful and important metric.

**Weaknesses:**

The authors make many unsubstantiated claims about human vision and search, without citation or reference to the abundant literature on these topics. There is a diverse set of opinions on how humans perform search tasks. E.g. Rosenholtz, et al.(2012). Rethinking the role of top-down attention in vision: Effects attributable to a lossy representation in peripheral vision. Frontiers in psychology, 3, 13., Carrasco, M. (2011). Visual attention: The past 25 years. Vision research, 51(13), 1484-1525., Wolfe, J. M., et al.(2017). Five factors that guide attention in visual search. Nature human behaviour, 1(3), 0058. None of the seminal work on understanding human attention is cited or even indirectly referenced in the paper. Specifically, there are no citations given to support that subjects in search tasks take an iterative, focus, rethinking, backtracking approach as claimed by the authors (e.g in lines 192-194). I don’t even find it necessary to connect improvements in VLMs at identifying camouflaged objects to inspirations in human perception. However, given the numerous claims in the paper including the title, the omission of citing research on human attention and real attempts at their application is unacceptable.

The human subject studies did not outline any steps towards the protection of the safety and privacy of subjects involved (e.g. IRB protocols, consent forms, etc.) As such I am flagging this for ethics review.

The human subject experiment is performed in LabelMe, a GUI for labeling images, without any mention of controlled stimulus presentation or timing as in a standard psychophysics experiment. How did the authors control the presentation time?

Prior work is not well introduced. It is only found in the appendix while typically, it should be near the beginning of the paper.

**Questions:**

There should be a border set of experiments to demonstrate generalizability of the VRRF framework. How does VRRF work on models other than Qwen2.5 and search tasks on other datasets?

How are the “Hard” vs “Easy” datasets identified? Was this an arbitrary grouping by authors of the paper or was it obtained through a human subject study based on accuracy, etc.?

A more detailed description of how training is performed is needed. For example, how is the prompt, q, presented to the various examples? Is it a fixed prompt as shown in figure 2? Or does the <explore> </explore> section vary between examples?

Minor points:
The clip art throughout the paper is distracting, and not appropriate style for a conference publication.
In Figure 8 the ordering 1243 is a bit confusing. It this incorrectly labeled? If not, it should be put in a raster pattern to avoid confusion.

---

### Official Review · Reviewer_nuQE · 2025-11-01

**Soundness:** 2
**Presentation:** 3
**Contribution:** 2
**Rating:** 4
**Confidence:** 4

**Summary:**

This paper proposes Visual Refocus Reinforcement Fine-Tuning (VRRF), an RL-based framework that teaches multimodal models to progressively “refocus” attention when detecting camouflaged objects. Using a modified GRPO algorithm and a curriculum-style reward schedule, the model learns human-like Focus–Rethink–Backtrace behavior, achieving large gains over SFT and even surpassing human accuracy on difficult camouflage datasets.

**Strengths:**

- Clear motivation: bridges a known gap between human and model perception in camouflaged scenes.
- Consistent improvement across COD datasets and competitive results versus GPT-4.

**Weaknesses:**

1. **Relative narrow scope**: The GRPO modification is simple and verified only on camouflaged perception. The paper would be stronger if it demonstrated generalization to broader perceptual or visual reasoning tasks.
2. **Experimental completeness**: (1) The “enhanced perception” claim should be better supported by results on perception-oriented benchmarks beyond camouflage (e.g., BLINK, CV-Bench, HallusionBench, or other perceptual / image captioning benchmarks). (2) It's not clear if the method focusing on camouflaged object will hurt general image understanding capability.
3. **Human-study details** are insufficient (sample size, time constraints, evaluation setup). The “surpasses human” claim should be qualified accordingly.

**Questions:**

- Can the designed mechanism generalize to other perception tasks (e.g. general / small object detection)? Additional numeric evaluations will be better here.
- Could the authors test whether the improved perception leads to measurable gains in overall model ability on general image understanding benchmarks (MMBench, MMVet)?
- What exactly is the role of "Clip-High Objective Without KL Penalty" in enhancing localization? Could it bias the model toward higher-confidence but less generalizable patterns? Evaluations for the last question can also demonstrate this.
- What is the backtracking behavior’s reliability? So far, it appears only in demo cases about camouflaged objects. Does it emerge consistently across different kind of VL tasks?

---

### Official Review · Reviewer_GedC · 2025-11-01

**Soundness:** 2
**Presentation:** 3
**Contribution:** 1
**Rating:** 4
**Confidence:** 3

**Summary:**

The paper proposes a RL-based framework to address the limitations of multimodal models in detecting camouflaged objects by aligning them with human visual perception. The method employs a progressive refocus mechanism, enabling models to iteratively refine attention and localize concealed objects through hierarchical reasoning. Using a combination of curriculum reinforcement learning and a rule-based reward system, the approach improves camouflaged object classification and detection. Extensive experiments on public benchmarks demonstrate significant performance improvements over SFT baselines.

**Strengths:**

- The paper is well-conducted.
- The experiments are solid.

**Weaknesses:**

- Given the context of COD, the use of bounding boxes instead of segmentation masks to indicate targets introduces ambiguity. From the perspective of reviewer, masks would provide a more precise representation of the target. Similarly, the comparison between human performance and the proposed method, particularly using mIoU as the metric, is invalid as human perception does not rely on bounding box-based localization (human use edge, boundary instead). Furthermore, the visualization results in Fig. 6 suggest that steps 2–4 do not provide significant improvements, merely adjusting the offset of bounding boxes.

- The rollout number. While the proposed method shows improved mIoU with increased reasoning steps, it is unclear whether the performance gain is due to the refocus mechanism itself or simply because additional reasoning steps allow for more rollouts. The increase in performance from RF1 through RF3 raises questions about whether these steps correspond to pass@1, pass@2, and pass@3, rather than reflecting a true reasoning improvement.

- Experiment details. Fig. 7 is not adequately explained in the paper. The meaning of RF1, RF2, RF3, RF4, and RF5 on the x-axis is unclear.

- The pipeline and proposed techniques bear significant resemblance to prior work, particularly DeepEyes. The approach appears to adapt general MLLM methodologies to the specific domain of COD without substantial innovation.

**Questions:**

see weakness

---

### Official Review · Reviewer_8yiG · 2025-11-01

**Soundness:** 3
**Presentation:** 3
**Contribution:** 3
**Rating:** 6
**Confidence:** 4

**Summary:**

This paper proposes Visual Refocus Reinforcement Fine-Tuning (VRRF), a reinforcement learning framework that aligns multimodal models with human camouflaged perception. By introducing curriculum-based rewards and in-context “refocus” trajectories, VRRF enables models to iteratively shift attention and reason from global to local regions, mimicking human visual focusing behavior. Experiments on multiple camouflaged object detection benchmarks show substantial gains over supervised fine-tuning and prior RL-based methods, even surpassing human performance in certain challenging settings. The work is novel, well-motivated, and demonstrates clear improvements, though it would benefit from comparisons with open-source reasoning models and deeper reward analysis.

**Strengths:**

- Aligning with human cognition is important because existing models only align the results and ignore the consistency of reasoning and cognition.
- The authors propose an innovative approach that utilizes visual grounding as a reinforcement learning method to enhance inference models.
- Construct a new benchmark.

**Weaknesses:**

- The understanding of ChatGPT in Figure 1 is not necessarily "misreading"; it may be due to the wolf's region (which may require an attribution-based approach to reveal), but it is weak in the ability to perform visual grounding based on text generation.
- The paper does not clearly specify whether rewards from previous stages remain active when transitioning to the next stage in the curriculum reinforcement learning process. It is unclear if the training is cumulative or reinitialized at each stage, which affects reproducibility and understanding of the optimization dynamics.
- The paper lacks comparisons with traditional camouflage object detection methods, which would better highlight the unique advantages of using MLLMs for camouflaged perception.
- The paper seems that do not evaluate the model’s generalization ability to out-of-distribution (OOD) camouflaged categories, leaving open whether the proposed refocus mechanism can handle unseen camouflage types or patterns.
- The paper does not discuss or visualize the model’s behavior when the refocus process fails to include the ground-truth region during reasoning. It remains unclear whether the model tends to produce incorrect predictions, hallucinate alternative explanations, or refuse to answer in such failure cases.
- It would be better to consider citing some works that differentiate between foreground and background for camouflaged object detection [1].
- This article may also discuss the importance of interpretability for visual grounding [2].

[1] Phantom-Insight: Adaptive Multi-cue Fusion for Video Camouflaged Object Detection with Multimodal LLM. 2025.

[2] Interpreting Object-level Foundation Models via Visual Precision Search. CVPR 2025.

**Questions:**

Please see the weaknesses.

---

### Note · Authors · 2025-11-13

**Comment:**

N/A

**Withdrawal Confirmation:**

I have read and agree with the venue's withdrawal policy on behalf of myself and my co-authors.